# Effect of Different Physical Therapy Interventions on Brain-Derived Neurotrophic Factor Levels in Chronic Musculoskeletal Pain Patients: A Systematic Review

**DOI:** 10.3390/life13010163

**Published:** 2023-01-05

**Authors:** Silvia Di-Bonaventura, Josué Fernández-Carnero, Luis Matesanz-García, Alberto Arribas-Romano, Andrea Polli, Raúl Ferrer-Peña

**Affiliations:** 1Escuela Internacional de Doctorado, Department of Physical Therapy, Occupational Therapy, Rehabilitation and Physical Medicine, Rey Juan Carlos University, 28933 Alcorcón, Spain; 2Department of Physical Therapy, Occupational Therapy, Rehabilitation and Physical Medicine, Rey Juan Carlos University, 28922 Alcorcón, Spain; 3Grupo Multidisciplinar de Investigación y Tratamiento del Dolor, Grupo de Excelencia Investigadora URJC-Banco de Santander, 28922 Madrid, Spain; 4La Paz Hospital Institute for Health Research, IdiPAZ, 28029 Madrid, Spain; 5Motion in Brains Research Group, Institute of Neuroscience and Movement Sciences (INCIMOV), Centro Superior de Estudios Universitarios La Salle, Universidad Autonóma de Madrid, 28023 Madrid, Spain; 6Grupo de Investigación de Dolor Musculoesqueletico y Control Motor, Universidad Europea de Madrid, 28670 Villaviciosa de Odón, Spain; 7Departamento de Fisioterapia, Facultad de Ciencias de la Salud, CSEU La Salle, Universidad Autonóma de Madrid, 28023 Madrid, Spain; 8Pain in Motion Research Group (PAIN), Department of Physiotherapy, Human Physiology and Anatomy, Vrije Universiteit Brussel, Pleinlaan 22, 1050 Brussels, Belgium; 9Centre for Environment and Health, Department of Public Health and Primary Care, KU Leuven, Blok D, Bus 7001, 3000 Leuven, Belgium

**Keywords:** pain, brain derived neurotrophic factor, physical therapy modalities, musculoskeletal chronic pain, electrotherapy, exercise, musculoskeletal manipulations

## Abstract

Objective: The main objectives of this review were, firstly, to study the effect of different physiotherapy interventions on BDNF levels, and, secondly, to analyze the influence of physiotherapy on pain levels to subsequently draw conclusions about its possible relationship with BDNF. Background: Based on the theory that neurotrophic factors such as BDNF play a fundamental role in the initiation and/or maintenance of hyperexcitability of central neurons in pain, it was hypothesized that the levels of this neurotrophic factor may be modified by the application of therapeutic interventions, favoring a reduction in pain intensity. Methods: A literature search of multiple electronic databases (Pubmed, PsycINFO, Medline (Ebsco), Scopus, WOS, Embase) was conducted to identify randomized control trials (RCTs) published without language restrictions up to and including March 2022. The search strategy was based on the combination of medical terms (Mesh) and keywords relating to the following concepts: “pain”, “chronic pain”, “brain derived neurotrophic factor”, “BDNF”, “physiotherapy”, and “physical therapy”. A total of seven papers were included. Results: There were two studies that showed statistically significant differences in pain intensity reduction and an increase in the BDNF levels that used therapies such as rTMS and EIMS in patients with chronic myofascial pain. However, the same conclusions cannot be drawn for the other physical therapies applied. Conclusions: rTMS and EIMS interventions achieved greater short-term reductions in pain intensity and increased BDNF over other types of interventions in chronic myofascial pain patients, as demonstrated by a moderate amount of evidence. In contrast, other types of physical therapy (PT) interventions did not appear to be more effective in decreasing pain intensity and increasing BDNF levels than placebo PT or minimal intervention, as a low amount of evidence was found.

## 1. Introduction

Chronic pain is a major global socioeconomic problem. According to a 2021 study, more than 30% of people worldwide are affected by chronic pain, making it one of the main causes of medical visits in primary care and of healthcare costs [1]. Until now, chronic pain was defined as a complex and multifactorial condition, usually triggered by tissue injury. However, nociception—the physiological processes that aim to detect and transmit potentially noxious stimuli from the tissues—is neither sufficient nor necessary for pain perception [2]. In fact, peripheral and central nociceptive pathways can become hyperexcitable in pain patients regardless of body site [3], making chronic pain not only a form of long-lasting acute pain, but a condition that, in some cases, may be associated with a sensitization of the nociceptive pathway and an alteration in biomarkers of plasticity that would not be explained by a simple pain–damage association [4].

Among the latter, neurotrophic factors have been proposed as therapeutic targets to improve the treatment of chronic pain [5]. One of the most important neurotrophins is brain-derived neurotrophic factor (BDNF), a polypeptide of the growth factor group, similar to neural growth factor (NGF) [6]. It is involved in multiple signaling learning and memory mechanisms [7], as well as a neuropathic pain biomarker [8]. In fact, BDNF plays a fundamental role in the initiation and/or maintenance of central nervous system hyperexcitability [9].

Specifically, the complexity of BDNF action depends on the isoforms in which it is present in the different areas of expression (i.e., CNS vs. periphery) [10]. In its mature version, it has two described actions that can be opposed, depending on the ligands to which it is linked. When this neurotrophin binds to TrkB receptors, it has a proactive role in the increase of neuroplasticity. On the contrary, when it binds to P-75 receptors, its functions are related to cell apoptosis and reduction in axonal processes, among others [10].

However, the exact role played by BDNF in modulating these signaling pathways during learning remains to be elucidated. Despite the limited knowledge about the role of this neurotrophin in the processing of nociceptive information, it appears to play a pro-nociceptive role and involved in the maintenance of hyperalgesic responses in patients presenting with chronic pain [7]. Therefore, the possible non-pharmacological manipulation of neurotrophic factors provides important new avenues of research. Physiotherapy in particular could be the first line of treatment in patients with persistent pain. Physiotherapy includes several approaches that are known to influence BDNF release, such as exercise [11], invasive (e.g., EIMS) [12], and non-invasive therapies (e.g., tDCS) [13]. The main objective of this review was to study the effect of different physiotherapy interventions on BDNF levels. Secondly, we analyzed the influence of physiotherapy on pain levels to subsequently draw conclusions about its possible relationship with BDNF.

Finally, the present study aimed to highlight the potential usefulness of other types of interventions, such as physiotherapy, in the non-pharmacological management of persistent pain.

## 2. Materials and Methods

This systematic review was conducted following the guidelines of the Cochrane Handbook for Systematic Reviews of Interventions [14], and the most recent guide, “Preferred Reporting Items for Systematic Reviews, PRISMA” [15]. The protocol was registered in the Open Science Framework with the DOI: https://doi.org/10.17605/OSF.IO/VA7TX, accessed on 21 July 2021.

### 2.1. Search Strategy

A systematic search for studies was conducted through the following databases: Pubmed, PsycINFO, Medline (Ebsco), Scopus, WOS and Embase, on 21 July 2021. An updated search was conducted in March 2022. The search strategy was based on the combination of medical terms (Mesh) and keywords relating to the following concepts: “pain”, “chronic pain”, “brain derived neurotrophic factor”, “BDNF”, “physiotherapy” and “physical therapy”. The complete search strategy is available in Appendix A.

### 2.2. Selection Criteria

#### Types of Studies

Only original studies of human clinical trials published without language restrictions were selected. Letters, posters or communications, abstracts, editorials, reviews, case series, and single-arm studies were excluded. No time restrictions were placed. Studies had to be controlled clinical trials. The selection criteria used in the present systematic review were based on clinical and methodological factors such as the Population, Intervention, Control, Outcomes, and Study design described by Stone et al. [16].

○PopulationAccording to these criteria, subjects with a chronic musculoskeletal pain condition were included and studies whose patients presented acute pain or neuropathic pain were excluded.○InterventionThe treatment had to be a physiotherapy intervention (using physical agents), including education and any modality of exercise and manual therapy was included.○ComparatorNo restrictions were placed on the comparator to analyze all the existing literature on the subject.○Outcome measuresIn each study, the peripheral evaluation of BDNF concentration (serum or plasma) before and after physical therapy treatment was performed along with perceived levels of pain.

### 2.3. Study Selection

The search duplicates were identified and removed. Unique articles were imported into the Rayyan application [17] to facilitate the screening. In the first stage, two authors [L.M.G. and A.A.R.] independently assessed the eligibility of the identified studies based on information from titles, abstracts, and keywords. During the second stage, the remaining full-text articles were independently reviewed for eligibility by both authors. A third author [J.F.C.] acted as a mediator if consensus was not reached.

### 2.4. Data Extraction

Data of the eligible studies were extracted by two independent reviewers [S.D.B. and R.F.P.]. A standardized data extraction of the following data was performed: study design, diagnosis, groups and their size, physiotherapy treatment, control group, sample and type of BDNF analysis, BDNF results, pain variable measured, and results on pain. Finally, both authors reached a consensus on each item of the extracted data. In the case of a disagreement between the authors, a third author made the final decision [J.F.C.].

### 2.5. Methodological Quality Assessment

The Physiotherapy Evidence Database (PEDro) scoring system was used to evaluate the selected studies. Two authors independently screened the full-text articles to obtain a score on the PEDro scale. The PEDro tool consists of 11 questions, with a maximum score of 10. The following criteria were used for rating the methodological quality of a study: nine to ten, “excellent”; six to eight, “good”; four to five, “fair”, and <four, “poor”. All studies were included in the analysis regardless of study quality [18].

Two independent reviewers (SDB and RFP) analyzed the quality of all selected articles using the same methodology. Disagreements between reviewers were resolved by consensus by including a third reviewer. Inter-evaluator reliability was determined using the “kappa coefficient” (>0.7 means high level of agreement among evaluators, 0.5–0.7 a moderate level of agreement, and <0.5 a low level of agreement) [19,20].

## 3. Results

### 3.1. Literature Search

The electronic search identified 865 studies. After reading the title and abstract, 749 were eliminated, leaving only 18 articles for full-text analysis. After the full-text screening, a total of seven studies were included in the qualitative analysis. The flow diagram is depicted in Figure 1.

### 3.2. Description of Included Studies

The description of the included studies is shown in Table 1. Three (43%) studies were conducted on subjects with chronic myofascial pain syndrome, two (28%) on subjects with knee osteoarthritis, and two (28%) on subjects with fibromyalgia. According to the intervention, two (28%) studies utilized exercise, one (14%) utilized transcranial direct current (tDCS), one (14%) utilized intramuscular electrical stimulation (EIMS), two (28%) utilized transcranial magnetic current (rTMS), and one (14%) utilized deep intramuscular electrical stimulation (DIMST). Five (71%) studies were compared with a placebo group and two (28%) were compared with a control group. Almost all studies used the Elisa kit and serum measurement for the analysis of BDNF levels. Only one [21] used the ECLIA test in plasma. For the pain variable, the majority (85%) of the studies used the visual analogical scale (VAS) [22], except for Liu’s study [23], which used the KOOS scale in a population with osteoarthritis of the knee.

### 3.3. Methodological Quality

The inter-rater reliability between the two independent evaluators on the PEDro scale was high (k = 0.897). The results for each of the studies are described in Table 2.

### 3.4. Effects of Physiotherapy on BDNF Levels

#### 3.4.1. Exercise

Significant changes in BDNF levels were not found in either in the resistance training group of Jablockhova et al., 2019 [21] in patients with fibromyalgia, in the study by Liu et al. [23] of Tai Chi, Baduanjin, or stationary cycling in patients with knee osteoarthritis.

#### 3.4.2. Electrotherapy

Both Dall’Agnol et al. [24] and Medeireos et al. [28] studied the effect of rTMS in patients with chronic pain. The former found a significant increase in the serum levels of BDNF when compared to pre-intervention. The latter study, despite having the same population and number of sessions, found no significant differences.

Da Graca-Tarragó et al. [27] found no evidence of changes in serum BDNF levels at the end of a tDCS treatment in people with knee osteoarthritis. In the case of dos Santos [26], there was a non-significant increase at the end of treatment in the intervention group.

In a population with chronic myofascial pain, an increase in the serum levels of BDNF was found after intervention with EIMS in the neck region [25]. However, no significant changes after the use of EIMS were found in people with knee osteoarthritis in the da Graca-Tarragó study [27].

### 3.5. Effects of Physiotherapy on Pain Levels

#### 3.5.1. Exercise

The Taichi, Baduanjin and cycling program decreased pain intensity (increased KOOS pain score) compared to a control of subjects treated with basic health education (Liu et al. [23]).

Jablockova [21], on the other hand, found no significant difference in pain levels measured with the visual analog scale after a resistance training protocol compared to relaxation therapy

#### 3.5.2. Electrotherapy

rTMS: rTMS showed a decrease in pain intensity measured with the VAS at the end of treatment compared to the placebo intervention [24]. Medeiros [28] also obtained significant positive results in pain reduction compared to the control group.

EIMS: Intramuscular electrical stimulation [25] showed statistically significant differences in pain levels compared to a placebo group.

tDCS: Da Graca-Tarragó [27] found differences in pain reduction with respect to the placebo group. In contrast, dos Santos [26] did not report the same result.

### 3.6. Relationship between Changes in BDNF Levels and Pain Severity

Two studies found a negative correlation between BDNF increase and pain intentionality (Dall’Agnol [24], Botelho [25]). On the contrary, Jablochkoba [21] did not find a significant correlation between the two variables. Da Graca-Tarragó [27] found that baseline BDNF levels correlated negatively with an increase in pressure pain thresholds. The studies of Liu et al. [23] and Medeiros [28] found a significant decrease in pain but no between-group differences in BDNF measurement. On the other hand, dos Santos did not present post-intervention results for either pain or BDNF.

Due to the heterogeneity of the interventions reported in the included studies and the small number of studies included, a metanalysis could not be carried out. Instead, we reported these findings in heatmaps (Table 3). Color coding was assigned according to the number of the studies reporting changes after the intervention.

## 4. Discussion

This is the first systematic review comparing the effect of different physical therapy interventions on serum BDNF and its correlation with pain levels. Seven studies with 373 participants were included. The main interventions were exercise and invasive and non-invasive electrotherapy. The reporting quality PEDro was collectively 8.14 points; the evidence seems to suggest a slight increase in serum BDNF concentrations after transcranial magnetic stimulation and intramuscular electrical stimulation, but not after other types of physiotherapy interventions. The assessment of pain in people with chronic musculoskeletal pain requires different parameters to be described. However, there is no consensus on which specific variables accurately describe the painful experience, and on the other hand, in the literature to date, the variable most used to describe pain is pain intensity, measured with the VAS or any of the similar unidimensional tools [29]. For this reason, in this review we have focused on the results of this variable and not others.

### 4.1. Exercise Therapy

Growth factors are a crucial mediator of the beneficial effects of exercise on brain function and pain perception [10]. According to the results of Lee at al. [30], it can be seen that BDNF concentrations in peripheral tissues of chronic pain patients seemed to increase following different exercise training protocols. However, within the included studies, the same cannot be said. Authors such as Liu et al. [23], who studied Tai Chi, Baduanjin and stationary cycling, seemed to observe no changes in BDNF levels but did observe changes in pain levels in people with knee osteoarthritis. On the other hand, Jablochkova [21] found no differences between strength exercise groups in either the BDNF or pain variables. Several factors may have influenced these results. First, different populations were compared: knee osteoarthritis and fibromyalgia. In the latter condition, it is common to find not only higher levels of brain neurotrophic factor and NGF, but also of pain threshold, making further analysis and comparison difficult [31]. Furthermore, the groups were compared with a control group that was offered relaxation therapy. It should be noted that BDNF has a negative correlation with cortisol [32], which could have biased the results of the study as a relaxation therapy was provided, and this may have influenced cortisol levels. The hypothesis of a down-regulation of BDNF by corticosteroids could be sustained if we consider the delay between BDNF mRNA expression and the subsequent process of translation, synthesis, and release of BDNF plasma protein [33]. From this point of view, it would be reasonable to suppose that high levels of cortisol may inhibit BDNF mRNA expression in the CNS. Finally, several considerations should be made before analyzing these results. First, exercise does not have an effect on BDNF directly, but on its TrkB receptor, meaning that as the treatment program progresses, the receptors increase in number, and in consequence BDNF is less present in the circulation because it has bound to the receptor [34,35]. Secondly, it is also interesting to note that depending on the number of weeks of exercise, different results were obtained. A single bout of exercise (sometimes referred to as acute exercise) has been shown to increase BDNF expression, and long-term exercise programs reduce (or normalize) BDNF [36], hence the possible lack of significant increases in this neurotrophin. 

Lastly, in Liu’s study [23], although more encouraging results were evidenced by the improvement in pain perception, the number of dropouts and other important methodological flaws, which made it impossible to carry out a reliable statistical correlation, must be considered.

### 4.2. Non-Invasive Neuromodulation Techniques: rTMS&tDCS

In the studies included in the present review, different results were obtained depending on the type of intervention provided (rTMS or tDCS) and the parameters of the single program.

#### 4.2.1. rTMS

For rTMS, Dall’Agnol et al. [24] found that 16 series of 10 s pulse/high-frequency of 10 Hz biphasic magnetic stimulator, with 26 s rest, significantly increased serum BDNF levels and reduced pain in patients with chronic myofascial pain syndrome. However, a slight change in the frequency and duration of rTMS, such as in the study of Medeiros et al. [28], can generate results that differ greatly from one study to another. In the latter study, therefore, no changes were obtained with the same type of population and number of sessions, but with 600 pulses at 10 Hz frequency and 80% resting motor threshold (rMT intensity).

#### 4.2.2. tDCS

In the case of tDCS, dos Santos et al., after applying an anode over the left DLPFC and the cathode at the right supraorbital region, using 2 mA for 20 min for 8 days in a population with fibromyalgia, did not obtain clear results. However, in patients with fibromyalgia, it is difficult to draw clear conclusions since they usually have altered levels of BDNF and other neurotrophins, as well as different pain processing variables, potentially as a result of high maladaptive plasticity, which makes their analysis very difficult. This theory would confirm what dos Santos [26] found: the effects of tDCS are partially dependent on initial BDNF levels. On the other hand, da Graca-Tarrago [24], with the application of an anode in the contralateral primary motor cortex (M1) and a cathode in the contralateral supraorbital region, at an intensity of 2 mA/5 s for 30 min, found no statistically significant differences with the control group. Due to the lack of existing literature on pain patients and their BDNF levels, we have not been able to compare with other trials.

In the case of invasive therapies such as EIMS and DIMS, contrasting results have been found when BDNF is considered. As far as the authors know, there is no literature that has studied the effect of these invasive therapies on BDNF levels in people with pain.

When da Graca-Tarrago et al. [27] compared EIMS in the lower limb with a sham stimulation, they found a reduction in pain levels after the intervention and a negative correlation of PPT with the initial BDNF levels. However, there were no significant changes in this neurotrophin at the end of treatment. Medeiros et al. [28], using DIMST, found a reduction in perceived pain levels but not in BDNF levels. These discrepancies could be due to the different populations and pathologies included in each report: elderly women with knee osteoarthritis, and with chronic myofascial pain, respectively. Although both studies were conducted in women, the age factor may be an important as the ability to produce BDNF decreases with age [37]. On the one hand, the influence of estrogens can alter BDNF [38] and its analysis. Another point to note with regard to their study is that, although they ruled out people taking corticosteroids, different patients were taking antidepressants [39], a drug that has been shown to alter the results of this neurotrophin and were overweight [40] according to baseline characteristics; these are all indicators that would alter BDNF expression. To conclude this group of invasive therapies, Botelho’s study was considered, in which he applied EIMS in the paraspinal region related to the nerve roots in people with myofascial pain. For both BDNF and pain, there were significant changes; BDNF showed an increase and pain a decrease.

Approximately half of the studies reported in this review did not find a significant change from the interventions in BDNF levels. Many factors may have contributed to this lack of great effect.

## 5. Limitations

This study had several limitations that affected data collection and qualitative analysis that should be considered. The first is that there was great heterogeneity because studies with different treatment protocols and therapies have been included, making it difficult to draw clear and homogeneous conclusions with so few studies included. The second limitation is that we have reviewed peer-reviewed articles without language restrictions, but only articles published in English with this characteristic were found, although there may be literature in other languages that we have not identified in our review. The third limitation is that the methods used to measure BDNF were different; most of the included studies measured it at the serum level, but one of the studies measured it in plasma form, thus producing heterogeneity in the results. This is in addition to being a secondary biomarker that can be altered by different stimuli that were put forward in the discussion and limitations of the study. The interpretation of the concentration of the BDNF levels could be hard to interpret since we cannot be sure of the origin of the biomarkers. It has been reported that BDNF measured in the peripheral nervous system could have a pro-nociceptive influence [8]. In contrast, the BDNF in the central nervous system could be a neuroplasticity marker. Most of the blood levels of BDNF come from the brain. However, BDNF can also be produced by endothelial cells or cells of the immune system [41,42,43]. The fourth limitation is that some of the included studies had a small sample size, and therefore these studies may have committed the statistical type II error (with an incorrect rejection of the null hypothesis). Moreover, no grey literature studies were identified that met the criteria for inclusion in the grey literature. Finally, we have not been able to perform a quantitative analysis with meta-analysis because we do not have enough data and articles to perform a subgroup analysis with sufficient methodological confidence.

## 6. Implications for Research

Several mediators may have contributed to the heterogeneity of peripheral BDNF levels in studies. In addition to the usual confounders such as circadian rhythm [44], menstrual cycle [45], baseline physical activity levels [46], and sex of the individual [47], the Val66met polymorphism has been shown to influence BDNF expression [48]. Furthermore, the pro- and mature forms of BDNF have opposite effects, in that pro-BDNF promotes cell death, while mature BDNF has the opposite function by promoting cell survival [49]. It should be considered in future studies how perceived pain intensity presents a statistically significant inverse correlation with BDNF in chronic myofascial pain syndrome subjects; however, in subjects with persistent pain, such as fibromyalgia and osteoarthritis, the correlation is different or directly positive. This, together with the fact that BDNF is a neurotrophin with a strong protective action on neuronal survival [49], as well as on neurogenesis, could be an indicator of maladaptive neuroplasticity in these type of chronic pain patients. This maladaptive neuroplasticity has been observed in the literature in subjects with similar characteristics [50]. In this sense, the possibility arises of finding therapies that can enhance the neurogenic action of the central nervous system, together with therapies that favor the correct alignment of this induced plasticity, to provide patients with persistent pain with a path towards the improvement of their quality of life and functional recovery. In addition, BDNF has been found to regulate nociceptive sensation and the plasticity of spinal dorsal horn neurons as a neurotransmitter and neuromodulator, contributing to the onset and progression of pain [51]. Other studies have found that noxious stimuli increase BDNF expression in the dorsal horn of the medulla [52]. Intratectal BDNF infiltration induces hyperalgesia and BDNF depletion reduces the degree of pain. BDNF is also involved in the regulation of plasticity of medullary GABAergic neurons [53].

## 7. Conclusions

In relation to the main objective of our study, we have observed that in the studies reported in the literature, BDNF levels after the application of different physiotherapy modalities have not varied in a statistically significant manner. More specifically, rTMS and EIMS seem to be more effective in temporarily elevating the levels of BDNF and reducing pain intensity than other strategies. Considering the importance of this theme, we cannot extract firm conclusions according to the results of this review. New studies are needed to investigate the types, intensities, and duration of physical therapy interventions that would be most appropriate to increase the peripheral levels of BDNF in people in pain.

## Figures and Tables

**Figure 1 life-13-00163-f001:**
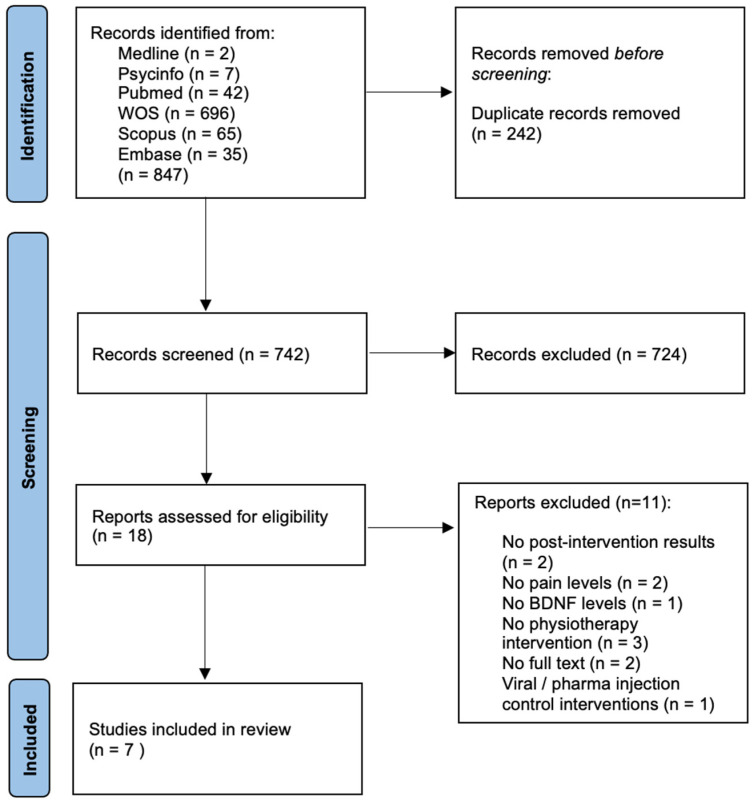
PRISMA 2020 Flow Diagram for new systematic reviews.

**Table 1 life-13-00163-t001:** Description of the included studies.

Author (Year)	Study Design	Diagnosis	Groups Characterization (Intervention: *n*, Age, % Females)	Intervention	Control	BDNF: Sample/Analysis Kit	BDNF Results	Pain Outcome	Pain Results
**Dall’Agnol et al., 2014 [24]**	RCT	CMPS	rTMS: *n* = 11, 45.83 (9.63), 100% Sham: *n* = 12, 44.83 (14.09), 100%	rTMS. Left motor cortex (M1). Trains: 16 series of 10 sec.pulse/high-frequency of 10 Hz biphasic magnetic stimulator 26 s rest. 1600 pulses per session. 10 sessions.	Sham Identical experience but without the impulses	Serum/ELISA	Not expose intra-group changes rTMS Increased BDNF vs. Sham	VAS	Higher pain score was correlated negatively with serum BDNF level [r-squared = 0.89, Beta= −0.15, SE = 0.008, (CI) 95% −0.17 to −0.13].
**Botelho et al., 2018 [25]**	RCT	CMPS	EIMS *n* = 12 48.36 (10.97) 100% Sham *n* = 12 46.00 (13.55) 100%	EIMS Paraspinal region related to the nerve roots (the splenius capitis and semispinalis capitis). 20-min at at 2 Hz. 10 sessions	Sham The same device but the output jack plug was broken	Serum/ELISA	Intra-group: EIMS increased BDNF EIMS Increased BDNF vs. Sham	VAS	Increase in serum BDNF induced by the EIMS was correlated negatively with pain at the end of follow-up (Beta: 0.67 t = 2.24 *p* = 0,02 (CI) 95% = 0.07 to 1.26)
**Dos Santos et al., 2018 [26]**	RCT	FM	tDCS *n* = 20, 49.15 (8.43) 100% Sham *n* = 20 50.05 (11.19) 100%	tDCS with a cognitive training task. Anode over the left DLPFC, cathode at right supraorbital region. 2 mA for 20 min. The cognitive training: online app of a Dual N-Back task. 8 consecutive days.	Sham The stimulator was turned off after a ramp-up of 30 s of stimulation	Serum/ELISA	Does not expose the results	VAS	Does not expose the results
**Da Graca-Tarragó et al., 2019 [27]**	RCT	Knee OA	a-EIMS/A-TDCS *n* = 15 66 (9.08) 100% A-tDCS/s-EIMS *n* = 15 64.4 (9.82) 100% S-TDCS/a-EIMS *n* = 15 64.40 (6.02) 100% s-tDCS/S-EIMS *n* = 15 63.87 (7.07) 100%	a-tDCS Anodal in the contralateral primary motor cortex (M1), cathode in the contralateral supraorbital region. 2 mA, rumps 30 s, 0.057 mA/cm^2^, electrodes 35 cm 2.5 s, 30 min. a-EIMS Needles (40 mm × 0.25 mm + constant current 2 Hz, intensity adjusted to tolerability, location 12 L1-S2, vast medial, rectur femoris, vast lateral, anterior tibialis and pes anserine bursae, 5 s, 30 min.	Sham transcranial direct current stimulation (s-tDCS) The stimulator was turned off after a ramp-up of 30 s of stimulation Sham intramuscular electrical stimulation (s-EIMS) electrodes were placed on the sites where the needles were placed. No electrical stimulation passed to the patient	Serum/ELISA	Not expose intra-group changes No difference between groups neither at baseline nor at the treatment end	VAS, PPT	Serum BDNF levels at baseline were correlated negatively with the PPT at the end of the treatment All groups showed decreased pain scores over time The aTDCS/aEIMS produced higher reduction compared with all three groups
**Medeiros et al., 2016 [28]**	RCT	CMPS	a-rTMS/a-DIMST *n* = 11 49.18 (11.63) 100% a-rTMS/s-DIMST *n* = 12 45.83 (9.63) 100% s-rTMS/a-DIMST *n* = 12 47.25 (11.00) 100% s-rTMS/s-DIMST *n* = 11 46.73 (13.09) 100%	a-TMS Coil over the left primary motor cortex. 600 pulses at 10 Hz frequency 80% resting motor threshold (rMT) intensity. 10 sessions for 20 min each one. a-DIMST Needles (40 mm × 0.25 mm). Nerve roots C2-C3, C3-C4, and C4-C5. Distance from the spinous process line: 1.5 cm). 10 sessions for 20 min using a frequency of 2 Hz.	Sham repetitive transcranial magnetic stimulation (s-TMS) A sham coil was used Sham deep intramuscular stimulation therapy (s-DIMST) Used an electroacupuncture device where the electrical connection between the stimulator and the patient was broken at the output jack plug of the stimulator	Serum/ELISA	No changes in BDNF levels	VAS	All groups presented lower level of Pain VAS than sham-rTMS sham-DIMST.
RCT: randomized controlled trial; rTMS: Repetitive transcranial magnetic stimulation; EIMS: Intramuscular electrical stimulation; DIMST: deep intramuscular stimulation therapy; tDCS: transcranidirect current stimulation; ELISA: enzyme-linked immunosorbent assay; BDNF: Brain derived neurotrophic factor; VAS: Visual Analogic Scale; KOOS: Knee injury and Osteoarthritis Outcome Score; PPT: Pressure pain threshold; FM: Fibromyalgia; CMPS: Chronic Myofascial Pain Syndrome: OA: Osteoarthritis; DLPFC: Dorso lateral prefrontal cortex.									
**Autor (year)**	**Study design**	**Diagnosis**	**Groups characterization (Intervention: *n*, age, % females)**	**Intervention**	**Control**	**BDNF: sample/analysis kit**	**BDNF Results**	**Pain outcome**	**Pain results**
**Liu et al., 2019 [22]**	RCT	Knee OA	**Exercise:** (a) Tai Chi *n* = 35 58.61 (7.62) (b) Baduanjin *n* = 35 59.66 (7.36) (c) Stationary cycling *n* = 35 61.26 (7.53) **Control** *n* = 35 56.88 (6.51)	**Exercise;** Tai Chi, Baduanjin and Stationary cycling. 1 h 5/w for 12 w	Control: Basic health education 1 h, 1/w for 12 w	Serum/ELISA	Not expose intra-group changes No difference between groups	KOOS pain	Taichi, Baduanjin and cycling decreased pain (increased KOOS pain score) vs. control
**Jablochkoba et al., 2019 [21]**	RCT	FM	**Resistance exercise***n* = 41 100% **Relaxation therapy** *n* = 34 100%	**Resistance exercise** Supervised: 10-min warm-up + 50 min. started with 40% MVC, developed up to 70–80%. 2/w for 15 w	Relaxation therapy: 25-min supervised mental exercises. 2/week.	Plasma/(ECLIA)	Intra-group: No changes in BDNF levels in FM (either in the exercise or in the relaxion group) No differences between groups (exercise vs. relaxion group)	VAS	No significant multivariate relationships were found between the changes in BDNF and pain in FM
RCT: randomized controlled trial; ECLIA: electrochemiluminescence assay panel; BDNF: Brain derived neurotrophic factor; VAS: Visual Analogic Scale; KOOS: Knee injury and Osteoarthritis Outcome Score; FM: Fibromyalgia; OA: Osteoarthritis.									

**Table 2 life-13-00163-t002:** Quality assessment of the included study according to the PEDro scale.

Evaluation Criteria	1	2	3	4	5	6	7	8	9	10	11	Total Score
Dall’Agnol et al., 2014 [24]	1	1	1	1	1	0	1	1	0	1	1	9
Botelho et al., 2018 [25]	1	1	1	1	1	0	1	1	1	1	1	10
Dos Santos et al., 2018 [26]	1	1	1	1	1	1	0	1	0	1	1	9
Da Graca-Tarragó et al., 2019 [27]	1	1	1	1	1	0	1	1	1	1	1	10
Medeiros et al., 2016 [28]	1	1	1	1	1	1	0	1	0	1	1	9
Liu et al., 2019 [22]	1	0	0	1	0	0	0	0	1	1	1	5
Jablochkoba et al., 2019 [21]	1	1	0	0	0	0	0	1	0	1	1	5

1: Specified Study Eligibility; 2: Random Allocation of Subjects; 3: Concealed Allocation; 4: Similarity Between Groups at Baseline; 5: Subject Blinding; 6: Therapist Blinding; 7: Assessor Blinding; 8: Fewer than 15% Dropouts; 9: Intention-to-Treat Analysis; 10: Between- Group Statistical Comparisons; 11: Point Measures and Variability Data.

**Table 3 life-13-00163-t003:** Heat map of the included studies.

Intervention	BDNF						Pain Intensity		
	SERUM			Plasma			VAS		
	Increase	Decrease	No Change	Increase	Decrease	No Change	Increase	Decrease	No Change
Exercise									
Electro-therapy									
*rTMS*									
*EIMS*									
*tDCS*									
*DIMST*									

## Data Availability

Not applicable.

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
