# Peer review of "Effect of Different Physical Therapy Interventions on Brain-Derived Neurotrophic Factor Levels in Chronic Musculoskeletal Pain Patients: A Systematic Review"

_life, 2023, doi:10.3390/life13010163_

Round 1

Reviewer 1 Report

The abstract of the article does not cite some of the databases that they claim to have used later in the study, although they do not provide detailed information on the results of each database. This information should be corrected in the abstract.
The study otherwise meets the scientific requirements to be a systematic review.
Review the change of font size in line 117.
The authors perfectly reflect all the limitations of their study.

Author Response

Dear editor,

We do appreciate the chance to review this manuscript. We have taken into account all of the reviewers' comments and feel that all of their concerns have been adequately addressed. We are grateful for the reviewers' helpful feedback, which has significantly improved the quality of our paper and made it easier to read. Attached is a list of revisions we have made, along with our responses to each of the reviewers' comments. We have highlighted all of the changes in the manuscript and included references to the specific comments in the text.

REVIEWER 1

Reviewer 1: The abstract of the article does not cite some of the databases that they claim to have used later in the study, although they do not provide detailed information on the results of each database. This information should be corrected in the abstract.

Response:

Dear Reviewer, thank you for your time and work to improve our manuscript.
Given the relevance of this aspect, we have decided to correct several things in the manuscript: first, we have redesigned the PRISMA Flow Diagram (line 175) by specifying the number of articles found in each database. Then we have detailed it in Appendix S1, and finally, we have rewritten the part of the abstract (line 44) and methodology (line 108) according to your suggestion.

Reviewer 1: Review the change of font size in line 117.

Response: Thank you very much for this correction (line 115-120), we have already fixed the paragraph and revised the others in case of possible similar errors.

Reviewer 2 Report

Very interesting manuscript, much appreciate the author's thoughtful approach and hard work. Some comments are as the following:

1. Please double check the database search is to Feb or March 2022?

2. Extensive editing of English language and style through the whole manuscript.

3. Modify the registration information in materials and methods section.

4. We know that Musculoskeletal pain affects bones, joints, ligaments, tendons or muscles. Did this review include all the eligible trials? How about the following three studies:

1)Gomes, Wellington F., Ana Cristina R. Lacerda, Vanessa A. Mendonça, Arthur N. Arrieiro, Sueli F. Fonseca, Mateus R. Amorim, Antônio L. Teixeira et al. "Effect of exercise on the plasma BDNF levels in elderly women with knee osteoarthritis." Rheumatology international 34, no. 6 (2014): 841-846.

2)Chassot M, Dussan-Sarria JA, Sehn FC, Deitos A, de Souza A, Vercelino R, Torres IL, Fregni F, Caumo W. Electroacupuncture analgesia is associated with increased serum brain-derived neurotrophic factor in chronic tension-type headache: a randomized, sham controlled, crossover trial. BMC complementary and alternative medicine. 2015 Dec;15(1):1-0.

3)Montero-Marin J, Andrés-Rodríguez L, Tops M, Luciano JV, Navarro-Gil M, Feliu-Soler A, López-del-Hoyo Y, Garcia-Campayo J. Effects of attachment-based compassion therapy (ABCT) on brain-derived neurotrophic factor and low-grade inflammation among fibromyalgia patients: A randomized controlled trial. Scientific Reports. 2019 Oct 30;9(1):1-4.

Author Response

Reviewer 2:  Please double check the database search is to Feb or March 2022?

Response: Dear Reviewer, thank you for your time and work to improve our manuscript.  Is it clear that there was a mistake when we wrote the date in line 109: it was realized in March. It has been correctly changed to avoid misinterpretations. We are sorry for the error.

Reviewer 2: Extensive editing of English language and style through the whole manuscript.

Response: We are aware of our language limitations and have decided to contact the MDPI English editor to improve the manuscript and its comprehension. The reference number of the revision of our manuscript is “MDPI English-57510” edited by Jonathan McCubbin. We will attach the English certificate of the journal along with the paper. Thank you very much for your suggestion!

Reviewer 2:  Modify the registration information in materials and methods section. very much.

Response: We are sorry for our error, and we have changed the wrong link of “OSF Registries”. It has also been modified in the materials and methods section (line 105) as: https://doi.org/10.17605/OSF.IO/VA7TX. Thank you very much for this comment.

Reviewer 2: We know that Musculoskeletal pain affects bones, joints, ligaments, tendons or muscles. Did this review include all the eligible trials? How about the following three studies:

1)Gomes, Wellington F., Ana Cristina R. Lacerda, Vanessa A. Mendonça, Arthur N. Arrieiro, Sueli F. Fonseca, Mateus R. Amorim, Antônio L. Teixeira et al. "Effect of exercise on the plasma BDNF levels in elderly women with knee osteoarthritis." Rheumatology international 34, no. 6 (2014): 841-846.

2)Chassot M, Dussan-Sarria JA, Sehn FC, Deitos A, de Souza A, Vercelino R, Torres IL, Fregni F, Caumo W. Electroacupuncture analgesia is associated with increased serum brain-derived neurotrophic factor in chronic tension-type headache: a randomized, sham controlled, crossover trial. BMC complementary and alternative medicine. 2015 Dec;15(1):1-0.

3)Montero-Marin J, Andrés-Rodríguez L, Tops M, Luciano JV, Navarro-Gil M, Feliu-Soler A, López-del-Hoyo Y, Garcia-Campayo J. Effects of attachment-based compassion therapy (ABCT) on brain-derived neurotrophic factor and low-grade inflammation among fibromyalgia patients: A randomized controlled trial. Scientific Reports. 2019 Oct 30;9(1):1-4.

Response: Thank you very much for the useful suggestion. We made the decision to exclude the first paper because it was a quasi-experimental study, which was not specified in the title. Initially, we had planned to read it in full, but upon realizing the design, we had to exclude it. On the other hand, we made the decision not to include electroacupuncture terms and psychology treatments in the literature search, as they were not related to physiotherapy interventions. As a result, the last two studies were not included in the search.

Round 2

Reviewer 2 Report

Accept!